# Diagnostics and Description of a New Subspecies of *Calluna vulgaris* (L.) Hull from Western Siberia

**Olga Cherepanova \*, Irina Petrova, Stanislav Sannikov and Yulia Mishchihina**

Botanical Garden, Ural Branch of the Russian Academy of Sciences, Ekaterinburg 620144, Russia;
irina.petrova@botgard.uran.ru (I.P.)
**\*** Correspondence: botgarden.olga@gmail.com

**Abstract:** The article presents the results of the study of fifty populations of common heather (*Calluna vulgaris* (L.) Hull) collected throughout its range. A phased comparative analysis (genetic, biochemical, anatomical, morphological, and ecological) was carried out with the estimation of indicators that included two key populations—Zavodouspenskoe (Pritobolye, Western Siberia) and Luga (Baltic, Eastern Europe). It was concluded that heather growing in Western Siberia should be identified as a separate taxonomic group, giving it the status of a subspecies. The gene pool of Pritobolye populations (including Zavodouspenskoe) is represented by the completely dominant (100%) monohaplotype S, which is not found anywhere else. The heather plant growing in Zavodouspenskoe has a longer lifespan. It is distinguished by larger linear leaf dimensions (length 2.06 ± 0.09 mm), thicker cuticle (4.77 ± 0.33 μm), increased number of trichomes (18.98 ± 0.56), and a reduced number of stomata (13.60 ± 0.63) than that growing in Luga. The new subspecies differs in biochemical composition: twice less content of epicatechin (average 1.992 ± 0.005 mg g$^{-1}$), three times more myricetin (average 2.975 ± 0.005 mg g$^{-1}$), twice as much chlorogenic acid (average 2.763 ± 0.004 mg g$^{-1}$). An ecological feature is that *C. vulgaris* does not grow in the swamps of Western Siberia and has a small population. This species has a high horticultural potential and requires protection as its population in Western Siberia continues to decline rapidly.

**Keywords:** heather; population; insular isolate; genetics; biochemical composition; anatomical and morphological features; coenoecology; divergence





## 1. Introduction

The shrub (*C. vulgaris*), which has always grown in the heathlands and damp places of northern Europe, is increasingly found in various types of gardens. Modern heather varieties vary in habit, corolla color, and foliage variability. Gardeners appreciate not only the attractive variability of heather, long flowering, and high plasticity but also the ability to easily tolerate pruning and bush formation.

For a long time, scientists considered heather to be the only species of the *Calluna* genus. No intraspecific taxones, except different morphoecotypes: *C. vulgaris* f. *alba*, *C. vulgaris* f. *aurea*, *C. vulgaris* f. *foxii*, *C. vulgaris* f. *searlei*, *C. vulgaris* f. *serotina*, *C. vulgaris* var. *hirsuta* [1,2], have been distinguished in its structure yet. In our view, it is mainly due to narrow regionality and insufficient development of quantitative population-based and geographical approaches [3,4]. Heather, endowed with a wide range, contains features that help it exist both in swamps and in arid places. Heather populations are also noted far beyond their main range: New Zealand, Australia, and North America. Heather's colonization of these regions is the result of sailors and scientist expansion from the European territories [5,6]. Its specific xeromorphic features allowed it to gain a foothold in the harsh conditions of poor soils and open areas of Australia [7–9]. Some small populations growing sparsely in the preforest-steppe of Western Siberia can endure periods of drought and retain seed germination for a long time, forming their reserve in soil. In most cases,

seed germination begins only after the fire. Forest fire, burning the moss layer, opens the possibility for the development of small seedlings of heather seeds without competition for light, water, and mineral nutrition.

Pritobolye populations (Western Siberia), geographically isolated for many years during the Pleistocene, have genetic features that distinguish these populations from their nearest neighbors from the main part of the continuous range (Figure 1). Genetic features are supported by a number of small but important ecological, anatomical, and biochemical features. Geographic isolation confirms the originality of heather growth in the region of the Tobol and Iset rivers [10–12].

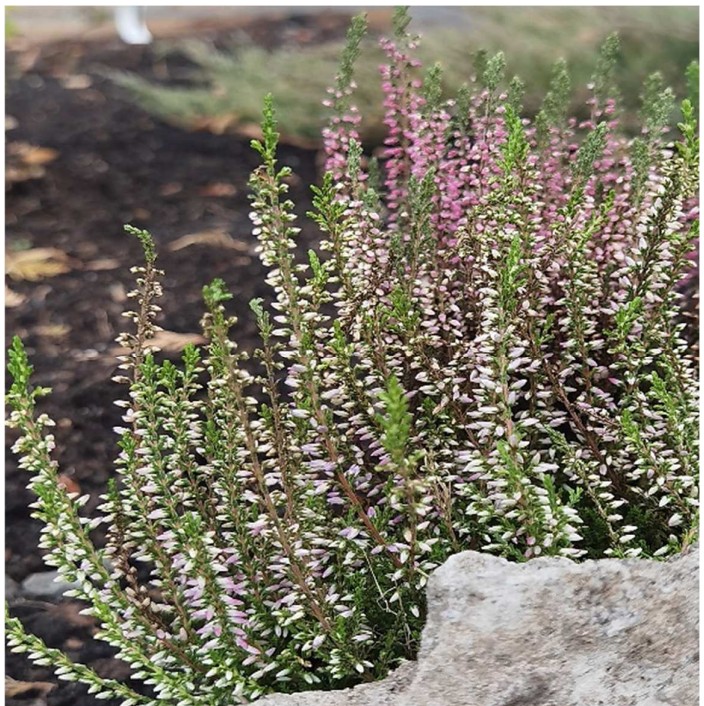

**Figure 1.** Young heather plant in Botanical Garden of Ural Branch of the Russian Academy of Sciences.

A high degree of xerophytization and a change in the species composition in the communities of the preforest-steppe of Western Siberia may be a consequence of the long-term determination of the aridity of the climate in this region. Heathers, despite their origin, were able to develop several adaptations that allow them to survive in the adverse conditions of hot summers. The low temperatures of the winter period in the Trans-Urals are easily tolerated by heather under a high snow cover. We attribute the fragmentation of the species range in Western Siberia to the lower frequency of fires than in European barrens and, importantly, to the lower soil temperature in winter. The leaf was able to adapt to the arid climate of Western Siberia by reducing the number of trichomes and stomata, as well as by a longer period of reduced transpiration. Low autumn and winter soil temperatures in Western Siberia are factors to which heather has no adaptation mechanisms. The marginal isolates, growing in extreme environmental conditions, namely protractedly isolated during Pleistocene relic, marginal, eastern, insular Pritobolye populations, located in the west of Western Siberia, are of great interest for revealing the process of intraspecific adaptive divergence of populations within the heather range. In the last ten years, versatile research of genetic, morphological, anatomic, and ecological peculiarities of the Pritobolye group of *C. vulgaris* populations was performed in the Botanical Garden of Ural Branch of the Russian Academy of Sciences (RAS) based on the ideas and approaches of Ural ecological-genetic science school of Schwartz–Timofeev-Resovsky [13–17]. The results of the quantitative analysis of structure parameters and geographic variation of Pritobolye populations, compared to other populations growing

within the entire species range, demonstrated significant differences between them, which enabled us to distinguish specific new taxon at the subspecies level—*C. vulgais* (L.) Hull ssp. *tobolica* [13]. Previously, a subspecies of *C. vulgaris* in Western Siberia was not distinguished.

Due to the low or declining numbers of these heather populations and the many threats they face, they have been assessed as critically endangered. Heather is listed in the Red Books of several regions of Russia [12]. The introduction of *C. vulgaris* representatives of these populations into culture can be considered as one of the steps towards their conservation. In general, the results obtained by the authors of this work were considered sufficient for the identification of a new specific taxon, the description of which is given below. The work aimed to clarify the ecological, morphological, anatomical, biochemical, and genetic differences between marginal populations and populations from the central part. The article summarizes, and our work is based on new population data and the parameters listed above.

## 2. Materials and Methods

The specificity of complex areographic, population-biological, genetic, anatomical, morphological, and coenoecological features of the Pritobolye group of *C. vulgaris* populations was grounded in applying the generic approach "species in the range".

As a result of long-term work (2009–2018), it was possible to carry out an analysis for fifty populations of heather, which were collected throughout the entire range (Figure 2 and Supplementary Table S1). For correct comparison, we chose two main populations in which ecological conditions are similar: Zavodouspenskoe (Pritobolye, Western Siberia) and Luga (Baltic, Eastern Europe). These two populations are present in various analyses and supplemented by others with similar ecological conditions to obtain the most reliable results.

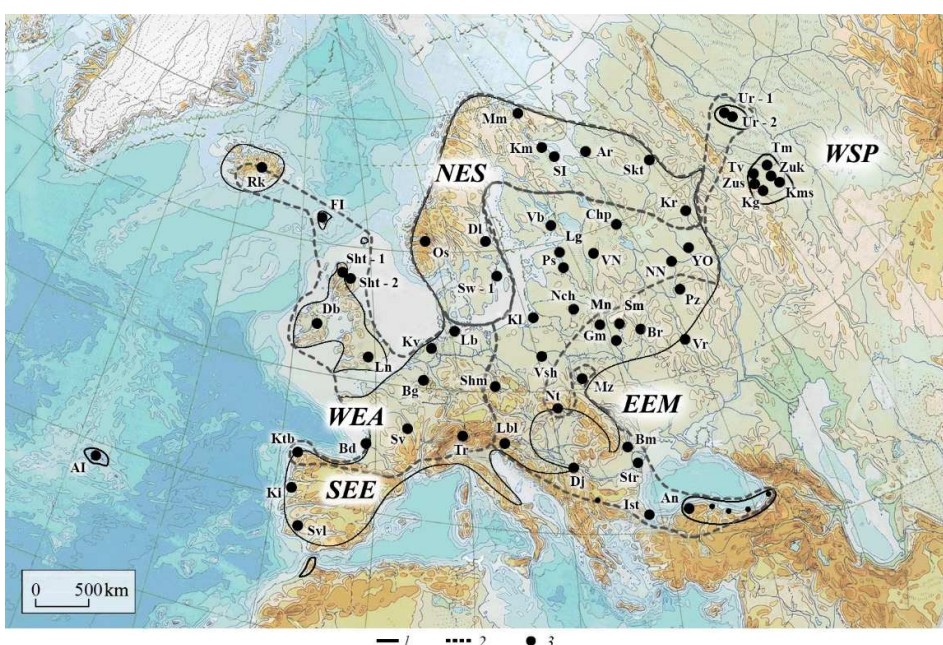

**Figure 2.** Geographic localization of population samples of *Calluna vulgaris* within the species range. 1—Range border of *C. vulgaris*, 2—Borders of phylogenogeographic regions, 3—Localization of *C. vulgaris* populations. NES—Northern-European-Scandinavian, WSP—Western-Siberian- Pritobolye, WEA—Western-European-Atlantic, EEM—Eastern-European-Mediterranean, SEE—Southern-Eastern-European.

## 2.1. Objects of Study

The central object of the study was a marginal, eastern group of insular populations, protractedly isolated during the Pleistocene from the main European *C. vulgaris* range and

located in Pritobolye, southwest of Western Siberia, Russia (Figure 2). Its contemporary range included sandy above flood-plain terraces of the Tobol River and its confluence [17]. Throughout the Pleistocene (over 1.8 million years), we think that the Pritobolye heather populations were reproductively and migratory separated from European populations. Glaciers and cold soils in the region of the Ural ranges and foothills have become the cause of isolation for heat-loving heather [15]. Insular populations of heather grew here, sometimes dominating, in the substage of Pritobolye pine forests on the dry sandy soil of cowberry-heather-green-moss pine forests and sometimes in other types of forests.

The second population of *C. vulgaris* (Luga) is located in the western part of the continuous range of the Russian Plain of the southern taiga subzone in the middle part of the slope of the Baltic sandy ridge in Eastern Europe. The material was collected in the lingonberry-heather-green moss pine forest. The soil is sandy-podzolic, and groundwater occurs at a depth of about 1.5 m. The stand is 110 years old; composition—10 C, absolute density 19.8 $m^2$ $ha^{-1}$, class II. Pine undergrowth, 15–20 thousand ind. $ha^{-1}$; undergrowth is absent. The moss layer (projective cover is about 70–75%) is dominated by the pleurocium (*Pleurozium schreberi*), occasionally dicranum (*Dicranum* sp); lichens of the genus *Cladonia*— 10%. In some places, there is a dead cover of up to 20%. Heather (*C. vulgaris*)—29.8% and lingonberry (*Vaccinium vitis-idaea*)—12% predominate in the grass-shrub layer; sedge (*Carex ericetorum*) occurs singly.

Population codes: AI—Azorian insels, Ar—Arkhangelsk, Bd—Bordeaux, Br—Bryansk, Bs—Belis, Chp—Cherepovets, Cv—Cevennes, Db—Dublin, Dj—Djerdap, Dl—Dalarna, FI—Faeroes, Gm—Gomel, Ist—Istanbul, Kg—Kurgan, Kl—Kaliningrad, Ki—Koimbra, Km—Kem, Kms—Komissarovo, Kr—Kirov, Kv—Katwijk, Lb—Lueneburg, Lg—Luga, Lbl— Ljubljana, Ln—London, Mn—Minsk, Mm—Murmansk, Mz—Mizun, Nch—Naroch, NN— Nizhny Novgorod, Nt—Nízke Tatry, Os—Oslo, Pz—Penza, Ps—Pskov, Rk—Reykjavik, Sm—Smolensk, Svl—Seville, Sw-1—Sweden-1, Shm—Šumava, Tv—Tavda, Tm—Tyumen, Tr—Trentino, Trb—Trabzon, Ur-1—Uraj-1, Ur-2—Uraj-2, Vb—Vyborg, VN—Veliky Novgorod, Vr—Voronezh, YO—Yoshkar-Ola, Zuk—Zavodoukovsk, Zus—Zavodouspenskoe, Vsh—Warsaw.

*2.2. Genetic Analysis*

The comparative analysis of the structure and differentiation of chloroplast DNA parameters was conducted for 6 insular Pritobolye populations on the one hand and 50 local populations from the rest part of the range on the other hand (Figure 2 and Supplementary Table S1).

The chloroplast DNA (maternally inherited in heather) was extracted from the leafy heather shoots taken from 8 to 37 samples in each population [18,19] and analyzed using the PCR-RFLP method [3,20]. The universal chloroplast primers were used, which turned out to be polymorphic in European populations of heather [21]. All these fragments were sequenced with the help of a sequencer (Applied Biosystem-3130, St. Louis, MO, USA) and edited using the software BioEdit [21]. Totally, 1500 heather specimens were analyzed in 50 local populations. The parameters $F_{ST}$ of the intra-group and inter-group variation were estimated by ANOVA based on the haplotype frequencies and the number of mutations in them, followed by the $N_{ST}$ parameter [22].

The results of this study are based on the ordination of all studied populations in a three-dimensional coordinate field, a result of hierarchical analysis, and a comparative quantitative assessment of genetic distances was given (average $F_{ST}$ values were determined).

*2.3. Morphological and Anatomical Analysis*

Comparative analyses and evaluation of morphological and anatomical differences were carried out for two previously selected populations: Zavodouspenskoe (Pritobolye, Western Siberia) and Luga (Baltic, Eastern Europe), located in the center of the species continental range (Figure 2). The main features of these relatively "standard" populations were compared under similar conditions of ecotopic and phytogenic environments—under the

canopy of Scots pine with close density (26.9 and 19.8 $m^2$ $ha^{-1}$, respectively) in geographically vicarious topoecologically similar types of green moss pine forests on sandy-podzolic soils [23]. To study the growth features of the heather root system, we dug up heather plants of various age groups growing on sandy substrates in Western Siberia under natural conditions.

Length and thickness of at least 5 leaves from the middle part of the three terminal annual leading shoots taken from 20 specimens from each population were measured. From the cuts of each leaf, fixed with a mixture of 95% ethanol and glycerol, we took 5 tissue samples for epidermis maceration, which was performed in NaOH water solution. The morphological and anatomical structure parameters of three to five leaves were measured for each specimen using the binocular loupe Carl Zeiss Stemi 2000-C (Carl Zeiss Meditec AG, Dresden, Germany), light microscope Carl Zeiss Scope A1 (Carl Zeiss Meditec AG, Dresden, Germany) and software Axio Vizion Rel. 4.8 (Carl Zeiss Meditec AG, Dresden, Germany). Main leaf features, representing the peculiarities of its adaptation to hydrothermal climate parameters, included the number and location of stomas and trichomes and the thickness of cuticle and epidermis.

The leaf elongation coefficient (LEC) was calculated by Equation (1):

$$LEC = L/Br, \tag{1}$$

where L—is the leaf length (mm), and Br—is the leaf width (mm).

## 2.4. Chemophenotypic Analysis

The bioactive substance content was assessed by applying reversed-phase HPLC with photometric detection. The extract of *C. vulgaris* was obtained according to the generally accepted method [24]: 1 g of crushed heather shoots was transferred into a flask in a water bath, 10 mL of 96% ethanol was added, and the mixture was extracted for 15 min. After that, the extract was poured through the filter paper into a dark container with a tight lid. Then 10 mL of 96% ethanol was again added to the residue. The procedure was repeated three times. The resulting extract was evaporated to an air dry state. The determination of the seasonal dynamics of the accumulation of biologically active substances was carried out by reverse-phase HPLC with photometric detection on an Aligent 1100 instrument (Agilent Technologies, Waldbronn, Germany) [25]. The following standards were used: chlorogenic acid, oleic acid, quercetin, myricetin, and epicatechin (Dia-m, Moscow, Russia). To prepare a standard sample with a concentration of 0.05 mg $cm^{-3}$, 5 mg of the standard was placed in a volumetric flask with a capacity of 100 $cm^3$, and the volume was brought to the mark with methanol. For analysis, the same extract as for determining the number of flavonoids was used.

Conditions for chromatographic analysis. Column: octadecyl silica gel 5 μm, 250 × 4.6 (e.g., 32 Phenomenex Luna 5 μm C18(2)); mobile phase: acetonitrile—trifluoroacetic acid solution pH 2.6 (40:60); mobile phase rate: 1.0 $cm^3$ $min^{-1}$; column temperature: 30 °C; detection: UV, λ = 365 nm. Injected sample volume: 10 $mm^3$. The calculation of the indicator components content (X) was carried out according to the calibration curve or according to Equation (2):

$$X = C \times S_1 \times V \times S_2 \times m, \tag{2}$$

where C is the concentration of the corresponding standard solution, mg $cm^{-3}$; $S_1$ is the peak area of the component in the analyzed sample; $S_2$ is the peak area of the component in the standard sample; V is the total volume of the sample dilution, $cm^3$; m—the mass of the sample.

## 2.5. Ecological Features

In natural communities, *C. vulgaris* is one of the typical species of pine forest underlayer [26]. The comparative analysis of such parameters as projective cover, growth, vitality, and ecologic range of heather populations was performed for different types of

geographically vicarious pine forests in Pritobolye (Zavodouspenskoe) and Baltic (Luga). As a result of "coenopopulation-based and microecosystemic" regression analysis, the same objects were used to compare the regional peculiarities of heather response to the tree stand, root, and light competition indexes [27].

## 2.6. Statistical Analysis

The results were analyzed using the Statistica 10.0 portable (StatSoft, Tulsa, OK, USA), Windows Excel, as well as specific programs Arlequin 3.5.1.2, Axio Vision Rel. 4.8 [28–31]. To assess the significance of the results, classical statistical parameters: mean value (M), mean standard error (SE), and coefficient of variation were used. For in-depth statistical analysis of nonparametric data, the ANOVA analysis (Statistica 10.0 portable) was used. The results of statistical treatment are given in the relevant sections. The results of the anatomical analysis were evaluated by several tests: Student's $t$-test and ANOVA. Significant differences are marked in the table with an asterisk.

## 3. Results and Discussion

### 3.1. Genetic Features

The most unbiased and systematic for the purpose of cross-regional population genetic differentiation assessment were the gene pool differences between their phylogeographic groups based on chloroplast DNA haplotype structure analysis in a 3-D coordinate space within the whole species range (Figure 3). The 3-D analysis evidenced for subdivision of the gene pool of 50 natural heather populations into five general phylogeographic groups, the boundaries of which are marked in Figure 2: Eastern-European-Mediterranean (EEM), Northern-European-Scandinavian (NES), Western-Siberian-Pritobolye (WSP), Southern-Eastern-European (SEE), Western-European-Atlantic (WEA) (Figure 3). The two populations that were compared belong to different groups: Zavodouspenskoe (Pritobolye) is included in WSP, while Luga (Baltic) is in SEE.

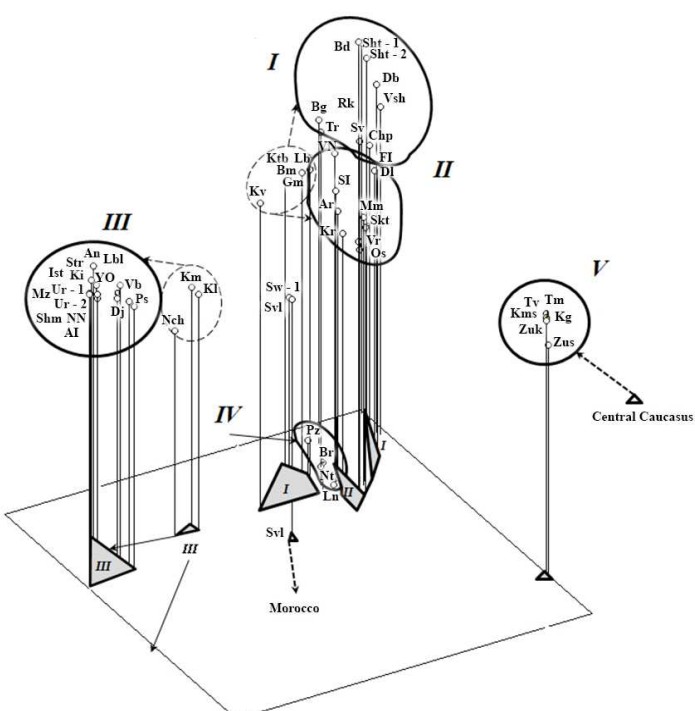

**Figure 3.** 3-D ordination of *C. vulgaris* populations. Population codes are given in Figure 2.

The most geographically and genetically separated population group of *C. vulgaris* included six marginal eastern protractedly isolated Pritobolye populations located in the southwest of Western Siberia. Their gene pool was represented by completely dominating

(100%) monohaplotype S, which could not be found anywhere else within the whole heather range. The origin of these haplotypes has been unknown yet, though we can assume that the Pleistocene refugium of Pritobolye heather populations, as well as of *P. sylvestris* L., was in Kazakhskiy Melkosopochnik where the relict heather location was noted [15,17]. The sharp genetic border of the Pritobolye group of populations—with complete replacement of monohaplotype S by monohaplotype D—can be noticed even in relation to the nearest populations of the northern Zauralye (Uraj, Figure 2).

The calculation of heather populations' intergroup genetic differentiation was carried out based on population hierarchical analysis performed on the whole species level (Table 1).

**Table 1.** Genetic distances ($F_{ST}$) between groups of *C. vulgaris* populations.

| Phylogenogeographic Groups of Populations | EEM | NES | WSP | SEE | WEA |
|---|---|---|---|---|---|
| Eastern-European-Mediterranean (EEM) | 0.00000 | | | | |
| Northern-European-Scandinavian (NES) | 0.60004 | 0.00000 | | | |
| Western-Siberian-Pritobolye (WSP) | 0.86929 | 0.63591 | 0.00000 | | |
| Southern-Eastern-European (SEE) | 0.43626 | 0.45641 | 0.90038 | 0.00000 | |
| Western-European-Atlantic (WEA) | 0.53501 | 0.19462 | 0.64500 | 0.47764 | 0.00000 |

Max mean genetic divergence was noticed between WSP, protractedly isolated in Pleistocene, marginal, eastern phylogeographic group of populations, and all the other groups, especially from the neighboring EEM (0.870) and SEE (0.901), which were isolated in Pleistocene by the vast Ural disjunction. The smallest pairwise intergroup distance was noted for WEA-NES (0.195), which indicates the free penetration of genetic flows between these close groups. We observed a decrease in the pairwise intergroup distance between population groups of the Iberian Peninsula (SEE) and populations of the southern Mediterranean (EEM), populations of the Iberian Peninsula, and populations of the eastern Atlantic (WEA).

*3.2. Morphological Features*

Table 2 shows the results of the comparative assessment of *C. vulgaris* vegetative shoots and leaves average morphoanatomical features. The analysis was carried out for typical Pritobolye (Zavodouspenskoe) and continental Baltic (Luga) populations.

The pivotal differential morphological feature of the Pritobolye typical *C. vulgaris* population, which allowed us to taxonomically distinguish it from the typical Baltic population of this species, was the linear dimensions of the leaves. Their average length with high reliability ($p \leq 0.00001$) differed 1.6 times from the European population. At the same time, the average thickness of leaves in the Western Siberian heather population was twofold more than in the typical Eastern European population (Table 2). The difference in leaf elongation coefficient between Pritobolye and Baltic populations was also significant. We did not note significant differences in the vascular system. Each leaf has one vein, along which the stomata are arranged in dense groups. The vein is displaced towards the center of the leaf when the leaf margins are closed, which reduces transpiration. There are less significant differences in the length and annual growth of the apical shoot, as well as in life expectancy and leaf growth, which were mainly determined by the stage of ontogenesis and interspecific competition.

**Table 2.** Leaf morphological and anatomical parameters of *C. vulgaris* in typical Pritobolye (Zavodous-penskoe) and Baltic (Luga) populations.

| Parameter | | Typical *C. vulgaris* Populations | | | |
| --- | --- | --- | --- | --- | --- |
| | | Zavodouspenskoe | Luga | $T_{ST}$ | *p* * |
| | | Morphological features | | | |
| Leaf length, mm | | $2.06 \pm 0.09$ [1] | $1.26 \pm 0.03$ | $-18.2810$ | 0.00001 |
| Leaf thickness, mm | | $0.76 \pm 0.02$ | $0.37 \pm 0.03$ | $-10.9994$ | 0.00001 |
| Leaf elongation coefficient (LEC) | | $0.39 \pm 0.02$ | $0.30 \pm 0.02$ | 2.0930 | 0.05 |
| | | Anatomical features | | | |
| | Stoma number, pc. | $13.60 \pm 0.63$ | $19.69 \pm 1.14$ | 4.3012 | 0.0002 |
| | Trichome number, pc. | $18.98 \pm 0.56$ | $15.65 \pm 0.75$ | $-2.8235$ | 0.008 |
| | Cuticle thickness, µm | $4.77 \pm 0.33$ | $3.97 \pm 0.28$ | 2.024 | 0.06 |
| | Height, µm | $38.08 \pm 2.56$ | $26.61 \pm 1.44$ | 2.069 | 0.05 |
| | Length, µm | $56.54 \pm 1.56$ | $58.78 \pm 2.4$ | 2.035 | 0.03 |
| Upper epidermis cells | Thickness, µm | $25.51 \pm 1.22$ | $29.39 \pm 4.46$ | 2.447 | 0.05 |
| | Area, µm$^2$ | $1126.43 \pm 61.40$ | $1195.19 \pm 56.37$ | 1.991 | 0.06 |
| | Perimeter, µm | $166.37 \pm 5.23$ | $180.48 \pm 5.02$ | 2.035 | 0.03 |
| | LEC | $0.453 \pm 0.01$ | $0.490 \pm 0.06$ | 2.018 | 0.05 |
| | Length, µm | $32.02 \pm 0.23$ | $26.19 \pm 1.14$ | $-4.9125$ | 0.00004 |
| | Thickness, µm | $13.82 \pm 0.05$ | $11.82 \pm 0.55$ | $-3.5992$ | 0.001 |
| Palisade tissue cells | Area, µm$^2$ | $402.25 \pm 8.50$ | $290.85 \pm 19.87$ | $-3.9669$ | 0.0005 |
| | Perimeter, µm | $112.23 \pm 7.99$ | $67.99 \pm 4.98$ | $-2.7204$ | 0.01 |
| | LEC | $0.432 \pm 0.003$ | $0.460 \pm 0.03$ | 2.101 | 0.05 |
| | Length, µm | $20.62 \pm 0.44$ | $16.76 \pm 1.06$ | $-2.8758$ | 0.007 |
| | Thickness, µm | $15.12 \pm 0.29$ | $13.29 \pm 1.48$ | 1.973 | 0.05 |
| Spongy parenchyma cells | Area, µm$^2$ | $252.68 \pm 6.96$ | $192.74 \pm 17.48$ | $-3.1006$ | 0.004 |
| | Perimeter, µm | $59.19 \pm 0.89$ | $54.99 \pm 3.25$ | 1.977 | 0.06 |
| | LEC | $0.738 \pm 0.01$ | $0.86 \pm 0.15$ | 2.011 | 0.05 |

[1] Data is presented as M ± SE. * Significance level of differences between studied *C. vulgaris* populations.

### 3.3. Morphological and Anatomical Features

Works devoted to heather leaves anatomical structure study are rare. Heather has a specific leaf structure, characteristic of xerophytic plants, which reduces the variability of traits. The small size of the heather leaf makes it difficult to study [28,29,31]. Throughout its range, heather exhibits weak plasticity, which may be due to the high specialization of the leaf. Due to a combination of anatomical and ecophysiological features, heather could be referred to as mesoxerophytes [11,32–34].

As the climate continentality in Zauralye increased, the xeromorphic features in leaf anatomical structure intensified, compared to heather growing in the European part of the range. It is true that the number of stomas bordering the channel on the lower side of the leaf decreased by 30%, $p \geq 0.0002$ (Table 2), and on the upper epidermis, there were almost no stomas (Figure 4), or they were very rare. The trichomes number increased (1.2 times), as well as epidermis cell height (1.4 times), with a simultaneous decrease in their length and thickness. Moving deeper into the continent of Eurasia in the eastern direction, heather plants are forced to adapt to the increasing dryness of atmospheric air, which is expressed in an increase in cuticular wax productivity (Table 2, Figure 4). An interesting feature is observed as we move deeper into the continent—a decrease in the tortuosity of epidermal cells. This can be seen from the decrease in the perimeter of the cell with relatively stable linear parameters of the epidermal cells. In Zavodouspenskoe, palisade chlorenchyma cells are larger than in Luga. At the same time, the elongation of the cell does not differ significantly, which indicates balanced changes in the elements. The area of spongy parenchyma cells is 1.3 times larger in Zavodouspenskoe than in Luga. The remaining linear dimensions of this type of cell do not change. The growth of mesenchymal cell linear dimensions led to the overall leaf linear dimensions rise. In general, almost all studied parameters specifying the sizes and forms of heather palisade parenchyma and

lacunose parenchyma in Pritobolye isolates differed from the ones in Baltic populations (Table 2).

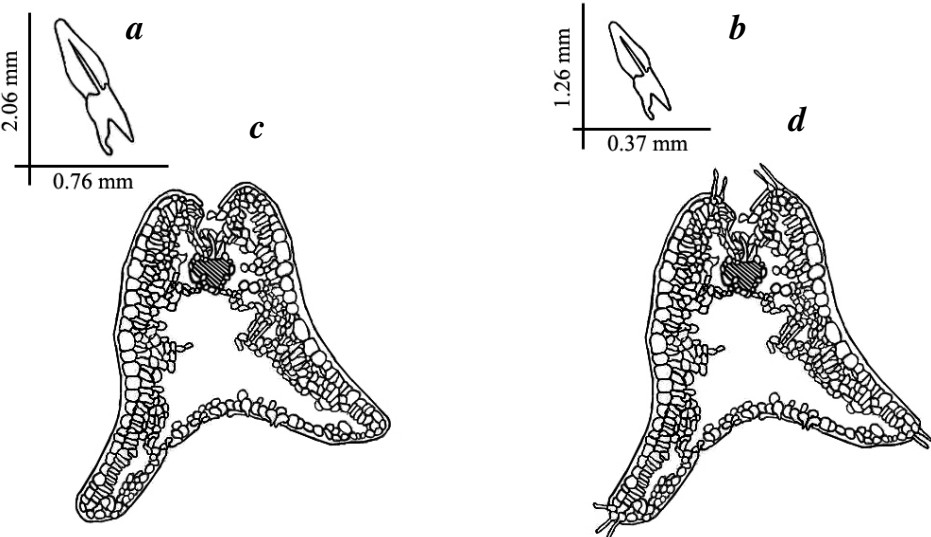

**Figure 4.** Schematic *C. vulgaris* leaf morphologic structure and cross-section view: (**a**,**c**)—Pritobolye type; (**b**,**d**)—Baltic type.

The statistically significant characteristics that form the special structure of the Pritobolye heather leaf include the height of the upper epidermis cells, the perimeter of the palisade tissue cells, and the area of spongy cells of parenchyma (based on Student's *t*-test). The statistical discrepancies found during the evaluation of the morphological and anatomical features using the Student criterion were confirmed with nonparametric tests for two independent samples (Mann–Whitney).

### 3.4. Chemotypic Analysis

Heather contains many biologically active components, allowing for expanding its application scope. We have focused on the most revealing components found in the green parts of the plant [35].

The chlorogenic acid and myricetin content in the tissues of the Pritobolye heather is significantly higher (by 2–8 times), and epicatechin is two times lower than in the European part of the range (Table 3). There is an increase in the content of almost all components, except for epicatechin, in the direction from west to east of the range. This can be associated with an increase in continentality, which is expressed in increased adaptability (possibly, an increase in the activity of the photosynthetic apparatus) to more severe environmental conditions. Thus, the chemotypic test could be used as an additional diagnostic method to prove the specificity of features of the distinguished taxon *C. vulgaris* (L.) Hull ssp. *tobolica* Sannikov.

**Table 3.** Biochemical composition of *C. vulgaris* extract.

| Biologically Active Substances, mg g$^{-1}$ | Typical *C. vulgaris* Populations | |
|---|---|---|
| | Zavodouspenskoe | Luga |
| Chlorogenic acid | 2.614 ± 0.005 [1] | 1.911 ± 0.005 * |
| Oleic acid | 0.198 ± 0.005 | 0.208 ± 0.004 |
| Quercetin | 2.982 ± 0.004 | 1.194 ± 0.005 * |
| Myricetin | 2.361 ± 0.005 | 0.451 ± 0.005 * |
| Epicatechin | 2.004 ± 0.004 | 2.954 ± 0.004 |

[1] Data is presented as M ± SE. * Significant differences between the studied *C. vulgaris* populations at $p \leq 0.01$.

*3.5. Ecologic Range*

The most pronounced and reliable coenoecologic differences between the Pritobolye Western-Siberian (Zavodouspenskoe) and Baltic Eastern-European (Luga) groups of *C. vulgaris* populations were determined in peculiarities of their regional topoecologic ranges. With the advancement to the east deep into the continent of Eurasia, heather is more often found under the canopy of pine forests and on the outskirts of raised bogs, avoiding flowing mesotrophic bog complexes. Its range is becoming more and more scattered, largely reflecting the range of pine forests growing on sandy terraces in Western Siberia [10–17].

In a generalized profile of topoecologically similar types of Pritobolye pine forests, the maximum heather projective cover in the cowberry-heather-green moss pine forest (30.4%) was almost the same as in a geographically vicarious type of forest in the Baltics (29.8%) (Figure 5). However, in the edaphic "dry" Pritobolye lichen pine forest, this parameter (7.5%) was 2.5 times less, and in the "wet" polytric pine forest (3.5%), contrariwise, this was 2.5 times more than in the Baltics (18.8 and 1.7%, respectively) [12,36].

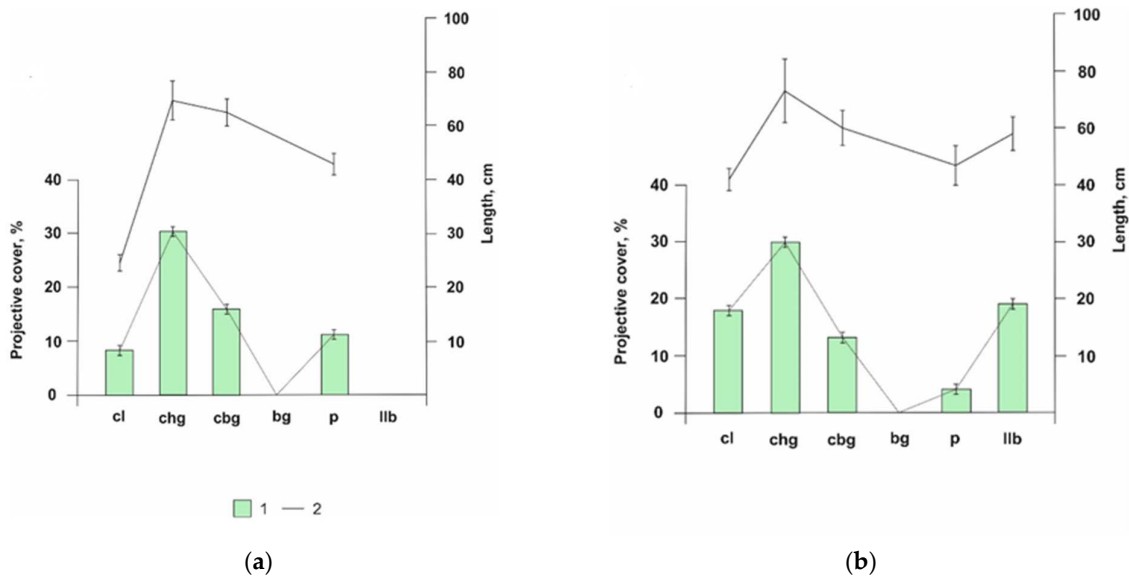

(**a**)                              (**b**)

**Figure 5.** Average projective cover and *C. vulgaris* shoot length in geographically vicarious types of pine forests of (**a**) Pritobolye, Western Siberia, and (**b**) Baltic, Eastern Europe. 1—Projective cover; 2—Leading shoot length. Data is presented as M ± SE. Types of pine forests: cl—cowberry-lichen; chg—cowberry-heather-green moss; cbg—cowberry-bilberry-green moss; bg—bilberry-green moss; p—polytric; llg—ledum-leatherleaf-bog moss.

*C. vulgaris* growing in Western Siberia, adapted to the increased dryness of atmospheric air and severe winter temperatures, forms a spherical shape of a bush with sufficient illumination (Figure 6).

Summarizing our results, we can identify some adaptive traits for heather from Pritobolye. Probably, heather mastered the territories of Western Siberia, dispersing from the regions of Northern Kazakhstan [4,14]. As dry conditions increase, the leaf reduces the number of stomata and trichomes and increases the thickness of the cuticle, which we consider to be an adaptation to decreasing humidity. Most often, heather growing in the main part of its range (Meadows) tends to open areas. It shows higher heliophilicity than heather in the Pritobolye. The accumulation of biologically active substances in plants depends on many factors. The concentration of biologically active substances can change during ontogenesis or under the pressure of environmental conditions. Thus, the content of secondary metabolites changes during the growing season and strongly depends on the altitude of the population, illumination, transpiration level, and the quality of water

exchange. For example, the amount of synthesized quercetin depends on the number of chloroplasts in the cell [11].

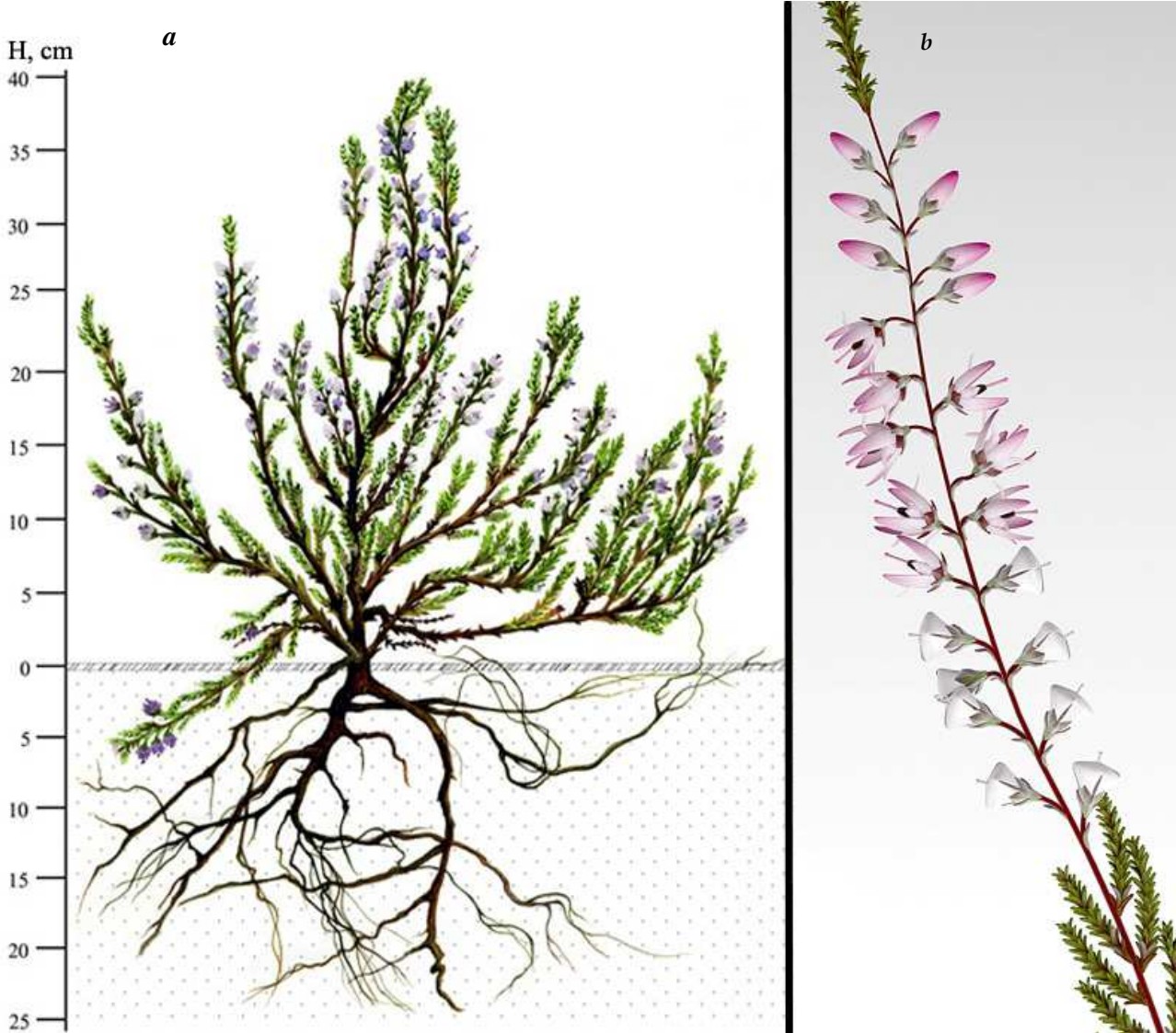

**Figure 6.** General view of (**a**) *C. vulgaris* 20-years-old adult (origin from seed) growing in Pritobolye, Western Siberia (Zavodouspenskoe); (**b**) Enlarged part of an annual shoot with leaves and inflorescences (3D- visualization).

Heather leaf anatomy was studied in European populations in the 20th century, but with only a descriptive purpose [33,34,37]. There were no comparable analyzes of heather leaves population variability. There are a few works describing the high adaptability of heather due to changes in soil conditions and the accumulation of heavy metals [38]. Heather growing in Western Siberia has a greater thickness of leaves than heather growing in the Russian Plain [17]. Most likely, the increase in the content of quercetin is associated with an increase in the number of chloroplasts in leaf cells, the number of which, in turn, also rises with an increase in the leaf linear dimensions, which changes following illumination decrease in the lower forest layer.

Botanic description of the typical full-grown specimen of *C. vulgaris* (L.) Hull ssp. *tobolica* Sannikov (tribe *Ericeae*; subfamily *Ericoideae*; family *Ericaceae*; order *Ericales* [39]) Evergreen semifrutex (Figure 6) of about 50–60 cm height growing under the canopy of pine tree stands (*P. sylvestris*), maximum 90 cm height on the open site. Shoots are intensely

branched, mostly lodging, heavy pubescent being young, shoot leafage—from 2 to 4 years. The leaves are imbricated in four rows, triangular small of 1.0–3.5 mm in length, squamose, downy, sagittal at the bottom, and sessile, with two awl-shaped outgrowths (Figure 6a,b). The stomas are mostly located on the abaxial side of the leaf in the channel and are covered with tight rows of trichomes, but there are almost no stomas on the adaxial side, or they are very rare. The flowers are gathered in a paniculate inflorescence consisting of falsely-racemose branches ending with foliated heads; sepals and lobes are quadripartite and membranous; the color of lobes varies from pale pink to purple (Figure 6b). The corolla is shorter than the cups and remains on the fruits. The number of stamens is 8, and the pistil has a four-blade snout and protrudes from the cup. The fruit represents a dry pubescent small boll (up to 1.5 mm in length). The seeds are small (0.8–1.2 mm in length), and the color varies from light brown to almost black.

Typus: herbarium sample is stored in the herbarium department of the Institute of Plant and Animal Ecology of Ural Branch of the Russian Academy of Sciences. International acronym of herbarium collection SVER0922190 (https://herbarium.ipae.uran.ru/about.html, accessed on 15 May 2018).

Distribution: The endemic scattered patches populations of heather growing in the river Tobol region on sandy terraces in the southwest of Western Siberia.

Similar Species: *Calluna vulgaris* (L.) Hull: a small shrubby plant usually growing less than 50 cm tall. Its tiny scale-like leaves are arranged in four vertical rows along the branches; it is pink or pale purple, bell-shaped, flowers have four petals that are joined together at the base, its tiny round capsules are hairy and have four compartments (Salisbury, Trans. Linn. Soc. London. 6: 317. 1802).

## 4. Conclusions

As a result of the comprehensive study of *Calluna vulgaris* populations in Western Siberia, significant differences were established between them and populations in all other parts of the species range. Our results are consistent with S. Wright's island model of microevolutionary divergence.

The most pronounced differences between the typical Pritobolye *C. vulgaris* population (Zavodouspenskoe) and continental East-European population (Luga) were found in the chloroplast DNA structure (revealing unique S haplotype) as well as in leaf linear dimensions (twice bigger) and xeromorphic features rise, which probably evidenced for adaptation to dry climate of Pritobolie, Western Siberia [12,13]. The divergence in other anatomic leaf parameters as well as in chemotypic features (bioactive substances content) in shoot tissues was less obvious. We revealed some alternate differences in the coenoecologic heather range in Pritobolye, where the heather was completely absent on peat moss bogs, unlike the European range.

Heather Pritobolsky can be recommended for breeding more drought-resistant varieties with high endurance, compactness of the vegetative part, a good growth rate of annual shoots, and a high rate of vegetative offspring formation.

In general, obtained research results are sufficient to distinguish specific taxon *C. vulgaris* (L.) Hull ssp. *tobolica* Sannikov in the infrastructure of *C. vulgaris* species. This species has a high horticultural potential and requires protection as its population in Western Siberia continues to decline rapidly.

**Supplementary Materials:** The following are available online at https://www.mdpi.com/article/10.3390/horticulturae9030386/s1. Table S1: List of fifty studied populations of *C. vulgaris* with their geographic coordinates.

**Author Contributions:** Conceptualization S.S. and I.P.; methodology, O.C.; software, O.C.; validation S.S. and I.P.; formal analysis O.C.; writing—original draft preparation, data curation, resources, Y.M.; writing—review and editing, O.C. and I.P.; visualization O.C.; supervision S.S.; project administration S.S.; funding acquisition I.P. All authors have read and agreed to the published version of the manuscript.

**Funding:** The work was carried out within the framework of the state task of the Botanical Garden of the Ural Branch of the Russian Academy of Sciences.

**Data Availability Statement:** All results are included within the article.

**Acknowledgments:** The authors express their thanks to L. Paule (Technical University in Zvolen, Slovakia) for collecting and providing the number of *C. vulgaris* samples from Europe.

**Conflicts of Interest:** The authors declare no conflict of interest.

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
