# Peer review of "Diagnostics and Description of a New Subspecies of Calluna vulgaris (L.) Hull from Western Siberia"

_horticulturae, doi:10.3390/horticulturae9030386_

Round 1
Reviewer 1 Report
Dear authors,
I let you down here some observation about your manuscript, after read it carefully:
- First of all, the introduction is ambiguous, and please improve it with other reference titles published in the field of your manuscript. Some statements presented in this section are not supported by bibliographic titles.
- What means "Tab CD"? (line 148)
- ANOVA is the correct name of the test, and not AMOVA (line 152)
- Figure 3 needs a better resolution, because is unreadable and difficult to understand it
- Please explain better the results presented in Table 1. A little large explanation is necessary.
- Last, but not the least, please improve the list of references. A number of just 28 titles is not acceptable for an international and high rated journal. More than that, some references are listed, but not inserted in the text (reference 5 for instance).
Overall, I can give a positive review after these major changes are done.
Best regards!
Reviewer 2 Report
The manuscript by Petrova et al. describes variation in traits among populations of Calluna vulgaris. The data set is large and comprehensive. The topic is appropriate for Horticulturae because of the horticultural appeal of the species, and because the horticulture industry poses a risk to the in situ populations as a result of harvesting. Overall, I really liked the manuscript. With appropriate revision the manuscript will be suitable for acceptance.
General:
1. The writing style is very dense and more technical than is needed. Many of the words are not common words and either should be changed to an alternative word or at least defined in the text.
2. A potentially fatal flaw of the manuscript is how the information adds to citation #8. If the distinction of the subspecies has already been published in the primary literature, why is this submission needed? The authors must address this head-on in the rebuttal and revision.
Specific:
Lines 27-28. This opening sentence does not even mention heather. Rewrite so it is a complete sentence.
Lines 39-40. The style of this sentence is literary, not scientific. Rewrite.
Line 59. Coenotic is not a word.
Lines 76-79. This sentence needs a citation.
Line 88. The sentence needs to be rewritten to explain what was done in citation 8 that differs from what is reported in the manuscript.
Line 91, 204, 325. Coenoecological is not a word.
Line 112 and other places. The authors inserted comments and the yellow highlights throughout the manuscript should have been deleted before submission. These must be deleted from the manuscript before resubmission.
Line 119 and other places. Every new binomial in the manuscript must have the taxonomic authority with the binomial the first time.
Line 217. AMOVA is not known by most biologists. Define this test here.
Line 252, Line 331. Spell out Maximum or define what Max means.
Line 267. Leaf elongation coefficient.
Line 275. Stand-edificator is not a word.
Line 286. Cuticle thickness cannot grow up. Please rewrite.
Lines 306-308. This general comment is not explained enough. Also, one sentence does not make a paragraph.
Line 313-314. You have a tense change. …was…was…is. Please fix.
Lines 320-321. One sentence does not make a paragraph.
Line 323, 340. Data are…; spell out mean.
Line 343. Figure 6 is not positioned near its first mention. Also, the methods used to excavate the root system should be added to the methods.
Line 350. Change “think” to “believe”
Lines 353-358. This general information needs a citation.
Line 379. Pansy is not a color.
Line 400. This statement needs a citation.
Line 411. Storoth is not a word.
Author Response
The authors are grateful for the detailed review of the manuscript.
We tried to take into account all the comments and highlighted them for you in blue.

Reviewer 3 Report
Authors have done a good job and presented their results well for presented topic. However, there is scope for further improvement before the manuscript is accepted for publication. Some important comments/suggestions are mentioned in pdf file for author’s attention and response.
Overall presentation of the manuscript is good. Language should be improved in many places by thoroughly reading the manuscript. Single sentences that make up a paragraph should be attached to other paragraphs.

Author Response
Dear Reviewer,
we have carefully reviewed the comments. And we tried to include everything. We highlighted the corrected fragments with color. We re-checked the manuscript. We tried to improve the language. In table 1, the abbreviations are deciphered in 1 column.
Figures 2 and 4 are completely drawn again.
Figures 5, 6 are original.
Thanks for your time and work.
